# Anti-Tumor Efficacy of In Situ Vaccination Using Bacterial Outer Membrane Vesicles

**DOI:** 10.3390/cancers15133328

**Published:** 2023-06-24

**Authors:** Elena Caproni, Riccardo Corbellari, Michele Tomasi, Samine J. Isaac, Silvia Tamburini, Ilaria Zanella, Martina Grigolato, Assunta Gagliardi, Mattia Benedet, Chiara Baraldi, Lorenzo Croia, Gabriele Di Lascio, Alvise Berti, Silvia Valensin, Erika Bellini, Matteo Parri, Alberto Grandi, Guido Grandi

**Affiliations:** 1Toscana Life Sciences Foundation, Via Fiorentina 1, 53100 Siena, Italy; e.caproni@toscanalifesciences.org (E.C.); a.gagliardi@toscanalifesciences.org (A.G.); m.benedet@toscanalifesciences.org (M.B.); g.dilascio@toscanalifesciences.org (G.D.L.); s.valensin@toscanalifesciences.org (S.V.); e.bellini@toscanalifesciences.org (E.B.); a.grandi@toscanalifesciences.org (A.G.); 2Department of Cellular, Computational and Integrative Biology (CIBIO), University of Trento, Via Sommarive 9, 38123 Trento, Italy; riccardo.corbellari@unitn.it (R.C.); michele.tomasi.2@unitn.it (M.T.); samine.isaac@benevolent.ai (S.J.I.); silvia.tamburini@unitn.it (S.T.); ilaria.zanella@unitn.it (I.Z.); marti.grigolato@gmail.com (M.G.); chiara.baraldi@studenti.unitn.it (C.B.); lorenzo.croia@unitn.it (L.C.); alvise.berti@unitn.it (A.B.); 3Department of Experimental and Clinical Biomedical Sciences “Mario Serio”, University of Florence, Viale Morgagni 50, 50134 Florence, Italy; matteo.parri@nikon.com; 4BiOMViS Srl, Via Fiorentina 1, 53100 Siena, Italy

**Keywords:** cancer vaccines, neo-epitopes, personalized medicine, outer membrane vesicles (OMVs), adjuvants

## Abstract

**Simple Summary:**

In situ vaccination (ISV) envisages the intratumoral injection of immunostimulatory molecules, which inflame the tumor and induce anti-tumor immune responses. Since bacterial outer membrane vesicles (OMVs) are naturally decorated with components which stimulate innate immunity, we tested whether OMVs can be used in ISV. Using three different tumor mouse models, we demonstrated the effectiveness of OMVs in inhibiting tumor development and in curing a large fraction of treated mice. We also show that if combined with tumor-specific neoantigens, the anti-tumor activity of OMVs is further enhanced. These latter results are particularly relevant since they support the use of a general strategy to optimize any in situ vaccination protocol. Considering their potency and the ease with which are produced, OMV-based ISV has the potential to become a standard of care for most solid tumors, particularly as a neoadjuvant therapy to be performed before surgery.

**Abstract:**

In situ vaccination (ISV) is a promising cancer immunotherapy strategy that consists of the intratumoral administration of immunostimulatory molecules (adjuvants). The rationale is that tumor antigens are abundant at the tumor site, and therefore, to elicit an effective anti-tumor immune response, all that is needed is an adjuvant, which can turn the immunosuppressive environment into an immunologically active one. Bacterial outer membrane vesicles (OMVs) are potent adjuvants since they contain several microbe-associated molecular patterns (MAMPs) naturally present in the outer membrane and in the periplasmic space of Gram-negative bacteria. Therefore, they appear particularly indicted for ISV. In this work, we first show that the OMVs from *E. coli* BL21(DE3)Δ60 strain promote a strong anti-tumor activity when intratumorally injected into the tumors of three different mouse models. Tumor inhibition correlates with a rapid infiltration of DCs and NK cells. We also show that the addition of neo-epitopes to OMVs synergizes with the vesicle adjuvanticity, as judged by a two-tumor mouse model. Overall, our data support the use of the OMVs in ISV and indicate that ISV efficacy can benefit from the addition of properly selected tumor-specific neo-antigens.

## 1. Introduction

Immunotherapy is one of the major breakthroughs in oncology, as demonstrated by the spectacular therapeutic efficacy of monoclonal antibodies, CAR T cells and adoptive transfer of tumor-infiltrating T cells [1,2,3,4,5,6].

Several other immunotherapeutic strategies are being developed, among which therapeutic cancer vaccines are emerging as a promising approach [7]. Recently, the development of efficacious cancer vaccines had been extremely challenging, mostly owing to the difficulties in identifying tumor-specific antigens and effective adjuvants [8]. Thanks to the discovery that mutations in cancer cells can create immunogenic epitopes (“neo-epitopes”) and that such neo-epitopes induce tumor-specific T cells, vaccines based on cancer neo-epitopes formulated with novel adjuvants have shown high efficacy in preclinical settings and have now reached the clinic with promising results, particularly in combination with other immunotherapies [9,10].

Neo-epitope-based cancer vaccines require the up-front identification of the tumor immunogenic mutations and are highly patient specific since most of the mutations vary from patient to patient. A different approach to cancer vaccination is in situ vaccination (ISV), which consists of the intratumoral administration of immunostimulatory molecules (adjuvants). ISV does not envisage the use of specific cancer antigens, the rationale being that antigens are already present at the tumor site and all that is needed is an adjuvant, which can turn the immunosuppressive environment into an immunologically active one. This strategy was conceptualized at the end of the nineteen century by Dr. William Coley. Observing several patients who fully recovered from cancer when an infection occurred at the surgical sites (a frequent outcome at that time), he developed a variety of strategies for treating cancers with live and dead bacteria or with bacterial extracts, later named “Coley’s Toxin”. He treated 896 patients, achieving five-year survival rates of 34% to 73% for inoperable carcinomas and 13% to 79% for inoperable sarcomas [11]. Although the use of the Coley’s toxin is no longer in the clinical practice, Dr. Coley’s pioneer work paved the way for the Food and Drug Administration (FDA) approval of the tuberculosis BCG vaccine as an ISV treatment of superficial bladder carcinoma and of the TLR7/8 agonist Resiquimod for skin carcinomas. In the January 2018–June 2021 period, 153 clinical trials dealing with the intratumoral injections of a variety of different formulations have been registered on ClinicalTrials.gov (accessed on 6 June 2021) [12].

ISV effectiveness relies on the potency of the immunostimulatory components delivered at the tumor site. Different adjuvants are being used, and they include CpG, Hiltonol, TLR4 agonists and agonists of the STING pathway [12]. As we previously anticipated [13], bacterial outer membrane vesicles (OMVs) have features particularly attractive for ISV applications. These vesicles, 30 to 300 nm in diameter, have a potent built-in adjuvanticity provided by a number of microbe-associated molecular patterns (MAMPs) naturally present in the outer membrane and in the periplasmic space of Gram-negative bacteria (LPS, lipoproteins, peptidoglycan, etc.). Such components trigger a potent inflammation and a Th1-skewed immune response, which ultimately promote cytotoxic T cell production and recruit phagocytic cells and NK cells. Moreover, the OMVs from several bacteria, including *E. coli*, induce immunogenic cancer cell death [14]. This feature is particularly interesting since the intratumoral OMV administration would favor the dissemination of cancer antigens, making them available for cross-presentation by DCs. Finally, OMVs can be decorated with foreign antigens and OMVs engineered with, or mixed to deliver cancer-specific peptide epitopes elicit anti-tumor immunity [15,16,17,18]. This offers the opportunity to test whether the inclusion of tumor-specific epitopes to the ISV formulation further potentiates the efficacy of ISV.

In this work, we first show that the OMVs from *E. coli* BL21(DE3)Δ60 strain, a strain deprived of several OMV endogenous proteins [19], promote a strong anti-tumor activity when intratumorally injected into the tumors of three different mouse models. Tumor inhibition correlates with a rapid infiltration of DCs and NK cells, occurring 24 h from the first OMV injection. We also show that the addition of five CT26 neo-epitopes to OMVs synergizes with the vesicle adjuvanticity, resulting in a potent anti-tumor activity, as judged by a two-tumor mouse model.

Overall, our data support the use of the OMVs in ISV and indicate that ISV efficacy can benefit from the addition of properly selected tumor-specific neo-antigens.

## 2. Materials and Methods

### 2.1. Bacterial Strain, Cell Lines and Mouse Strains

*E. coli* BL21(DE3)Δ60 was produced in our laboratory [16] and grown at 30 °C under shaking conditions (200 rpm).

The cell lines B16-F10, CT26 and WEHI-164 were from ATCC. All cell lines were cultured in RPMI supplemented with 10% FBS, penicillin/streptomycin/L-glutamine and grown at 37 °C in 5% CO_2_.

C57BL/6 or BALB/c female 8-week-old mice were purchased from Charles River Laboratories and kept and treated in accordance with the Italian policies on animal research at the animal facilities of Toscana Life Sciences, Siena, Italy and Department of Cellular, Computational and Integrative Biology (CIBIO)—University of Trento, Italy. Mice were caged in groups of 5/8 animals in ventilated cages.

### 2.2. Synthetic Peptides

The synthetic peptides M03 (DKPLRRNNSYTSYIMAICGMPLDSFRA), M20 (PLLPFYPPDEALEIGLELNSSALPPTE), M26 (VILPQAPSGPSYATYLQPAQAQML TPP), M27 (EHIHRAGGLFVADAIQVGFGRIGKHFW) and M68 (VTSIPSVSNALNWKEFSFIQSTLGYVA) were purchased from GeneScript (Piscataway, NJ, USA) in lyophilic form and solubilized in milliQ water at a final concentration of 5 mg/mL.

### 2.3. OMV Preparation

OMVs_Δ60_ were prepared growing the *E. coli* BL21(DE3)Δ60 strain in an EZ control bioreactor (Applikon Biotechnology, Schiedam, The Netherlands) until the end of the exponential phase at 30 °C, pH 6.8 (±0.2), dO_2_ > 30%, 280–500 rpm. OMVs were then purified and quantified as previously described [16]. Briefly, the culture supernatant was separated from biomass by centrifugation at 4000× *g* for 20 min. After filtration through a 0.22 μm pore size filter (Millipore, Burlington, MA, USA), OMVs were isolated, concentrated and diafiltrated from the supernatant using Tangential Flow Filtration (TFF) with a Cytiva Äkta Flux system (Marlborough, MA, USA). OMVs were quantified using DC protein assay (Bio-Rad, Hercules, CA, USA). OMV proteins were separated using a 4–12% gradient polyacrylamide gel (Invitrogen, Waltham, MA, USA) and finally stained with Coomassie Blue (Giotto, Sesto Fiorentino, Italy).

### 2.4. Dynamic Light Scattering Analysis

Size distribution profile of OMVs was determined by Dynamic Light Scattering (DLS) based on laser diffraction method using Zetasizer Nano (Malvern Panalytical, Malvern, UK). The OMV diameter of the batch preparation diluted at a final concentration of 0.5 mg/mL in PBS was determined by measuring the 90° side scatter size at 25 °C. Three measurements (between 15 and 20 experimental runs for each measurement) were averaged to determine the vesicle size.

### 2.5. Negative Staining Electron Microscopy Analysis

A volume of 5 μL of OMVs diluted at 80 ng/μL in PBS was loaded onto a copper 200-squaremesh grid of carbon/formvar rendered hydrophilic by glow discharge using a Q150R S (Quorum, Laughton, UK). The excess solution was blotted off after 30 s using Whatman filter Paper No.1 (Maidstone, Kent, UK). The grids were negatively stained with NanoW (Nanoprobes, Yaphank, NY, USA) for 30 s, then blotted using Whatman filter Paper No.1 and finally left to air dry. Micrographs were acquired using a G2 Spirit Transmission Electron Microscope (Tecnai, Dawson Creek, NE, USA) equipped with a CCD 2k × 4k camera at a final magnification of 120,000×.

### 2.6. Confocal Microscopy

CT26 cells (1.5 × 10^5^/well) were plated on microscope coverslips in 6 multi-wells plate (Corning, New York, NY, USA) and incubated at 37 °C o/n. Subsequently, labelled OMVs with 2 μM BODIPY™ 493/503 (ThermoFisher Scientific, Waltham, MA, USA) were incubated with cells at 37 °C for 24 h and washed twice with 1X sterile PBS. Nuclei were stained blue using DAPI (300 nM). Glass coverslip-plated cells were fixed with 3.7% formaldehyde solution in PBS for 20 min at 4 °C. Then, cells were permeabilized with 0.1% Triton X-100 (ThermoFisher Scientific, Waltham, MA, USA) in PBS, stained with 50 μg/mL of phalloidin-tetramethyl-rhodamine isothiocyanate (TRITC) for 1 h at room temperature (RT) to visualize F-actin and mounted with ProLong™ Gold Antifade Mountant (ThermoFisher Scientific, Waltham, MA, USA). All fluorescence samples were examined at RT using a laser-scanning confocal microscope TCS SP5 (Leica, Mannheim, Germany). Lasers and spectral detection bands were chosen for the optimal imaging of TRITC, and FITC PMT levels were set using control samples. Multicolor images were collected simultaneously in two channels. Images were taken using a 63×, 1.4 NA, oil, HCX Plan APO lens (Leica, Mannheim, Germany). Images were captured using the Leica LAS-AF image acquisition software 1.2. Overlays were generated using LAS-AF software 4.0.

### 2.7. Lactate Dehydrogenase (LDH) Release Cytotoxicity Assay

CT26 cells were seeded at 1 × 10^4^ cells per well in 100 μL of RPMI1640 without phenol red (Corning, New York, NY, USA), with 10% FBS (heat inactivated) (ThermoFisher Scientific, Waltham, MA, USA), 2 mM L-glutamine (EuroClone, Milan, Italy) and 1X penicillin/streptomycin (EuroClone, Milan, Italy), in a 96-well flat bottom, tissue culture-treated plate (Corning, New York, NY, USA) and then incubated overnight at 37 °C, 5% CO_2_. The next day, the culture medium was carefully removed so as not to disturb the cells. Treatments and controls were added in a final volume of 200 μL of treatment media (RPMI1640 without phenol red, 2 mM L-glutamine and 1X penicillin/streptomycin and 1.25% FBS). Negative control was treatment media only, positive control was Triton X-100 (ThermoFisher Scientific, Waltham, MA, USA) at 0.1%. OMVs, derived from BL21(DE3)ΔompA and BL21(DE3)Δ60, added in the following concentrations: 1 μg/well; 5 μg/well; 10 μg/well; 50 μg/well. Treated cells were incubated for 24 h at 37 °C 5% CO_2_. The plate was centrifuged for 10 min at 250× *g* and 100 μL of cell-free supernatant was transferred to a clear flat bottom 96-well plate. Subsequently, 100 μL of freshly prepared reaction mixture (catalyst and INT dye solution) was added and incubated for 30 min at RT in the dark. Then, the reaction was stopped with 1M HCl. The absorbance of colorimetric reaction was measured at 492 nm (reference wavelength: 620 nm) using an Infinite M200PRO Plate reader (TECAN, Männedorf, Swiss). The percentage cytotoxicity was calculated as follows: cytotoxicity % = (sample value-negative value)/(positive value-negative value) × 100.

### 2.8. Mouse Tumor Models

#### 2.8.1. One-Tumor Model

Eight-week-old C57BL/6 and BALB/c female mice were subcutaneously challenged in one flank with 1 × 10^5^ of B16-F10 and with 2.5 × 10^5^ CT26, or 1.5 × 10^5^ WEHI-164 cells, respectively. When tumor size reached a volume of approximately 100 mm^3^, 10 μg OMV_Δ60_ in 50 μL PBS, or PBS alone (50 μL), was injected into the tumors, and the treatment was repeated two additional times at two-day intervals. Tumor growth was followed for at least 30 days after the challenge and tumor volumes were determined with a caliper using the formula (A × B^2^)/2, where A is the largest and B is the smallest diameter of the tumor. The re-challenge experiment was performed by injecting 2.5 × 10^5^ CT26 cells into naïve BALB/c mice and in mice previously cured with OMVs_Δ60_. Tumor growth was followed as described before. Statistical analysis (unpaired, two-tailed Student’s *t*-test) and graphs were processed using GraphPad Prism 5.03 software.

#### 2.8.2. Two-Tumor Models

BALB/c female mice were 7 weeks old when the experiments began. A amount of 2.5 × 10^5^ CT26 colon carcinoma cells were injected subcutaneously at sites on both the right and left flank of the mouse. Tumor size was monitored every 2 days on both sides of the animals with a digital caliper and expressed as volume (volume = (width 2 × length)/2). When tumor size reached at ≥50–100 mm^3^, mice were vaccinated in situ in one of the tumors. Mice were immunized with PBS, OMVs_Δ60_ (10 μg/injection) or OMVs_Δ60_ adsorbed with the previously described five synthetic peptides (“pentatope”, 20 μg of each peptide). Mice were vaccinated three times every two days.

### 2.9. Flow Cytometry Analysis

For the analysis of different tumor cell populations, tumor-infiltrating lymphocytes were isolated from subcutaneous CT26 tumors as follows. Two tumors per group were collected and minced into pieces of 1–2 mm in diameter using a sterile scalpel, filtered using a Cell Strainer 70 mm (Miltenyi Biotech, Bergisch Gladbach, Germany) and transferred into 50 mL tubes. Then, the tumor tissue was enzymatically digested using the Tumor Dissociation kit (Miltenyi Biotech, Bergisch Gladbach, Germany) according to the manufacturer’s protocol, and the gentleMACS™ Dissociators (Miltenyi Biotech, Bergisch Gladbach, Germany) were used for the mechanical dissociation steps. After dissociation, the sample was passed through to a 30 mm filter (Miltenyi Biotech, Bergisch Gladbach, Germany) to remove larger particles from the single-cell suspension. At the end of the dissociation protocol, 1–2 × 10^6^ cells from tumors were incubated with Fixable Viability Stain UV440 (BD Bioscience, San Jose, CA, USA) for 15 min at RT in a 96-well plate. Then, cells were washed twice in PBS and incubated with 25 µL of an anti-mouse CD16/CD32-Fc/Block (BD Bioscience, San Jose, CA, USA) 20 min on ice in the dark. Later, 25 µL of the following mixture of fluorescent-labeled antibodies was added to the samples: CD3-APC (Biolegend, San Diego, CA, USA), CD4-BV510 (Biolegend, San Diego, CA, USA) and CD8a-PECF594 (BD Bioscience, San Jose, CA, USA) CD44-APC-Vio770 (Miltenyi Biotech, Bergisch Gladbach, Germany), CD25-PE-Vio770 (Miltenyi Biotech, Bergisch Gladbach, Germany), aNK1.1-PerCP-Vio770 (Miltenyi Biotech, Bergisch Gladbach, Germany), αMHC II (I-Ek)-VioBright (Miltenyi Biotech, Bergisch Gladbach, Germany), CD103-VioBright 515 (Miltenyi Biotech, Bergisch Gladbach, Germany) and CD62L-VioBlue (Miltenyi Biotech, Bergisch Gladbach, Germany). Cells were stained at RT in the dark for 20 min. After two washes with PBS, cells were fixed using the Fixation/Permeabilization Solution of FoxP3 Staining Buffer Set (Miltenyi Biotech, Bergisch Gladbach, Germany) for 20 min at RT, and cells were then washed twice and re-suspended in 50 µL of Anti-FoxP3-PE (Miltenyi Biotech, Bergisch Gladbach, Germany) diluted in permeabilization buffer and stained 20 min at RT. Cells were finally washed twice in permeabilization buffer and then in PBS. Samples were analyzed using a Symphony A3 (BD Bioscience, San Jose, CA, USA), and the raw data were elaborated using FlowJo software V10 6.1.

### 2.10. Immunofluorescence Assay

CT26 tumors were excised 24 h after a single intratumoral injection of 10 μg OMV_Δ60_ in 50 μL PBS, or PBS alone (50 μL). Tumors were embedded in Tissue-Tek^®^ O.C.T. (Leica, Mannheim, Germany) and rapidly frozen in dry ice. Cryosections of 8 μm in size were obtained from frozen tumors at Cryostat (Leica, Mannheim, Germany). CT26 tumors were stained with primary antibodies for rabbit anti-mouse Caspase 3 (Invitrogen, Waltham, MA, USA), rat anti-mouse NKp46 (Biolegend, San Diego, CA, USA), rat anti-mouse Dendritic Cell marker 33D1 (Biolegend, San Diego, CA, USA), and goat anti-mouse CD31 (R&D System, Minneapolis, MN, USA). Three secondary antibodies were used: donkey anti-rat IgG Alexa Fluor 488 (Invitrogen, Waltham, MA, USA), donkey anti-goat IgG Alexa Fluor 594 (Invitrogen, Waltham, MA, USA) and goat anti-rabbit IgG Alexa Fluor 488. DAPI (ThermoFisher Scientific, Waltham, MA, USA) was used to detect the nuclei. Stained sections were mounted with Dako fluorescence mounting medium (Agilent, Santa Clara, CA, USA) and examined with a Eclipse Ti2 microscope (Nikon, Minato, Tokyo, Japan). Images were analyzed with Fiji for Mac OS X software.

## 3. Results

### 3.1. OMVs_Δ60_ Are Internalized by Cancer Cells and Promote Cancer Cell Death

Vanaja et al. previously reported that OMVs from different Gram-negative bacteria, including *E. coli* BL21, promote the killing of cancer cells through Caspase-11-dependent pyroptosis [14]. More specifically, the authors showed that OMVs are efficiently internalized via endocytosis by different cell types, and the OMV-associated LPS is released into the cytosol from early endosomes, activating caspase-11. Activated caspase-11 not only triggers caspase-1 activation (together with NLRP3 and ASC) but also directly cleaves gasdermin D, thus mediating pyroptosis. The OMV-mediated killing of cancer cells is an interesting feature for ISV since the intratumoral injection of OMVs would favor the dissemination of cancer antigens in the tumor microenvironment.

Since our OMV vaccine platform is based on the use of vesicles from *E. coli* BL21(DE3) derivatives carrying a detoxified penta-acylated lipid A and the cumulative inactivation of proteins naturally present in OMVs [19], we first tested whether our OMVs were internalized by cancer cells and capable of promoting cell killing.

To this aim, OMVs_Δ60_ were purified by Tangential Flow Filtration (TFF) and the consistency, in terms of protein profile and size distribution, of the OMV preparation with that previously described [19] was confirmed by SDS-PAGE, Dynamic Light Scattering (DLS) and Electron Microscopy (EM) (Figure 1A–C). Next, CT26 were incubated with OMVs_Δ60_, and their binding to and internalization into cancer cells were followed by confocal microscopy. As shown in Figure 1D, in less than one hour of incubation, the vesicles accumulated on the cell surface, and in 24 h, approximately 50% of the visualized cells carried vesicles in their cytoplasmic compartment. Furthermore, we followed the killing of CT26 after the 24 h incubation with OMVs using the in vitro LDH release assay. As shown in Figure 1E, the vesicles promoted the killing of 18.2% ± 0.4 of the cells as opposed to 12.1% ± 0.6 killing observed with medium alone.

From these data, we concluded that similarly to that previously reported for other OMVs, OMVs_Δ60_ have a cytotoxic activity despite the fact that they carry a detoxified penta-acylated LPS.

### 3.2. In Situ Vaccination with OMVs_Δ60_ Inhibits Tumor Growth in Immunocompetent Mice

Next, we tested whether OMVs_Δ60_ could inhibit the growth of different tumor cells implanted s.c. in immunocompetent syngeneic mice. Three tumor mouse models were used: BALB/c mice challenged with CT26 colorectal cancer cells, BALB/c mice challenged with WEHI-164 sarcoma cancer cells and C57BL/6 challenged with B16-F10 melanoma cells. Mice were challenged s.c. with 2.5 × 10^5^, 1.5 × 10^5^ and 1 × 10^5^ cancer cells, respectively, and when the tumors reached a size of approximately 100 mm^3^, either PBS or 10 μg of OMVs_Δ60_ was injected intratumorally three times at two-day intervals. As shown in Figure 2 OMV administration substantially inhibited tumor development in all mouse models (Figure 2A,B), and 33% to 61% of mice became completely tumor free (Figure 2C). Interestingly, soon after the first OMV administration, necrotic areas were well visible in the tumors (Figure 2D).

To demonstrate the elicitation of tumor-specific immunity, two experiments were carried out. (i) In the first experiment, performed in two independent immunization rounds, seven of the BALB/c mice which were cured from the CT26 tumor were re-challenged s.c. with the same cell line 68 days after the first challenge. No tumor growth was observed in all OMV-vaccinated animals compared with naïve mice (Appendix A). (ii) In a second experiment, naïve BALB/c mice were challenged s.c. with 2.5 × 10^5^ CT26 tumor cells, and when tumors became palpable (approximately 100 mm^3^), 1 μg of OMVs_Δ60_ was injected intratumorally three times, two days apart. One day after the third injection, the tumors were collected and analyzed by flow cytometry to follow the presence of memory T cells. As shown in Appendix A, in OMV_Δ60_-treated mice, the substantial reduction in intratumoral live cells (Appendix A) was accompanied by an increase in the percentage of CD8+ central memory (CD44+/CD62L+) T cells (Appendix A).

To shed light on the inflammatory response taking place soon after the intratumoral injection of the OMVs, four animals from both the control group and the vaccinated group were sacrificed 24 h after receiving the first dose. Tumors were dissected, and the cellular population was analyzed by flow cytometry. As shown in Figure 3A, in line with the observed necrosis taking place at the tumor site, the tumors from vaccinated mice had a reduced number of live cells. Moreover, a higher infiltration of NK cells and MHC-II+/CD103+ DCs was found in the tumors of the OMV-treated animals with respect to the tumors from the control mice (Figure 3B,C). This suggests that the OMV-mediated inflammation rapidly creates an immunological environment which favors tumor cell killing and tumor antigen cross-presentation. As predicted, no substantial changes in CD3+ lymphocytes were observed after 24 h from vaccination.

### 3.3. Combination of OMVs_Δ60_ with Neo-Epitopes Improved the Efficacy of In Situ Vaccination in Two-Tumor Mouse Models

As already indicated, ISV is based on the administration of immunostimulatory molecules necessary to “inflame” the tumor and does not foresee the injection of tumor-specific antigens since such antigens are already available at the tumor site. However, it is now clear that among the plethora of tumor neo-epitopes present in the tumor microenvironment, only a few are immunogenic and have an anti-tumor activity [4]. Therefore, the in situ co-administration of immunostimulatory molecules and selected cancer neo-epitopes should optimize the anti-tumor immune responses, reducing the “diluting effect” of irrelevant epitopes.

To test this hypothesis, we used the more stringent two-tumor mouse model, according to which the animals are challenged with a cancer cell line at two distal sites, and when tumors become palpable (100 mm^3^), one tumor only is injected with either OMVs alone or with OMVs decorated with known tumor-specific neo-epitopes (Figure 4A). Since a potent tumor-specific immune response is needed to inhibit the distal, non-treated tumor (“abscopal” effect), the contribution of the neo-epitopes to the protective activity of OMVs can be appreciated by comparing the extent of tumor inhibition mediated by the two vaccine formulations.

OMVs_Δ60_ (10 μg) were combined with five synthetic peptides (“pentatope”), 20 μg each, corresponding to the sequences of five CT26-specific neo-epitopes. We previously showed that these neo-epitopes, originally described by Kreiter et al. [20], elicited epitope-specific T cells when injected s.c. in BALB/c mice together with OMVs_Δ60_ [16]. Moreover, and importantly, the epitopes protected BALB/c mice from CT26 challenge when systemically administered as synthetic RNA [20]. BALB/c mice were challenged with CT26 cells at two sites and “pentatope”-OMVs were injected into one palpable tumor of each mouse. The growth of both tumors was followed over a period of 26 days (Figure 4A). As shown in Figure 4B, the injection of OMVs_Δ60_ substantially delayed the development of the treated tumor, but the lateral tumor was in general marginally affected. By contrast, the combination of OMVs with the pentatope strongly inhibited both tumors.

## 4. Discussion

OMVs are emerging as a promising platform for vaccine development. Because of their ability to elicit a potent Th1-skewed immune response, OMVs are being used not only for prophylactic vaccines against infectious diseases but also for designing therapeutic cancer vaccines, the efficacy of which depends on the elicitation of tumor-specific cytotoxic CD8+ T cells and CD4+ Th1 cells. For instance, we previously showed that the s.c. immunization of immunocompetent mice with *E. coli* OMVs engineered with tumor-specific B and T cell epitopes inhibited tumor development [15,16]. Moreover, Kim et al. reported that the i.v. delivery of vesicles purified from both Gram-negative and Gram-positive bacteria resulted in an anti-tumor activity which correlated with the infiltration of T cells in the experimental tumors [21]. In addition, Cheng et al. chemically associated tumor T cell epitopes to *E. coli* OMVs and showed that such OMVs protected mice against the challenge with tumors expressing such epitopes [22]. Finally, OMVs expressing specific tumor neo-epitopes either released by intestinal commensal bacteria or administered by oral gavage were shown to protect mice from tumor challenge [18,23].

In this work, we further expand the applicability of OMVs to cancer therapy by demonstrating that their intratumoral injection has potent anti-tumor activity. Three different tumor mouse models were used, and in all of them, a relevant tumor-inhibitory effect was observed. Such an effect was mediated by the potent inflammation induced by OMV injection, which promoted a prompt infiltration of NK cells and DCs in the tumor microenvironment. The rapid migration of MHC-II+/CD103+ DCs appears particularly interesting since these cells are expected to internalize the tumor antigens released at the tumor site by dying cells and to promote the production of effector T cells in the draining lymph nodes. The accumulation of NK cells can partially explain the relevant tumor cell death observed in vaccinated mice. However, such killing is also mediated by OMVs, which have been shown to induce immunogenic cell death (pyroptosis), and our in vitro data confirm this observation.

As previously pointed out, ISV typically involves the intratumoral administration of immunostimulatory molecules (adjuvants) assuming that the relevant tumor antigens are readily available in the tumor microenvironment. This makes the therapy particularly attractive since, differently from personalized vaccines, it does not have to be adapted to patients. However, one interesting result emerging from our study is that OMV-based ISV is more effective if the injected formulation includes tumor neo-epitopes. Since s.c. immunization with OMVs decorated with neo-epitopes elicits epitope-specific T cells [16], the tumor microenvironment should be populated with a high concentration of selected effector T cells, which are expected to amplify the effect of the OMV-mediated inflammation at the tumor site.

It remains to be tested whether the tumor antigens have to be co-injected into the tumor or if they can be delivered separately. For instance, it would be interesting to combine personalized cancer vaccines with OMV-based ISV. Our expectation is that the two strategies synergize, enhancing the infiltration of T cells elicited by personalized vaccines into the tumor microenvironment.

In addition to combining OMV-based ISV with cancer vaccines, other combinations should be tested, including in situ vaccination and checkpoint inhibitors (anti-CTLA4 and anti-PD1/PDL1 mAbs). This combination is particularly attractive since if effective, lower doses of mAbs could be used, thus reducing toxicity and costs.

## 5. Conclusions

ISV offers several advantages over systemic immunotherapy strategies. First, it is difficult to imagine any better way to induce a potent inflammation within the tumor microenvironment. Such inflammation, taking place where cancer antigens have the maximal concentration, should guarantee the highest possible polyclonal production of anti-tumor T cells, which can reach metastatic sites via systemic circulation (“Abscopal” effect). Second, ISV should mitigate the typical (often severe) adverse reactions of systemically delivered formulations by virtue of the fact that the active substances are administered at a lower dosage exactly where needed (avoiding off-target toxicity).

The effectiveness of ISV relies on the optimal selection of immunostimulatory components. Several PRR agonists are under evaluation in clinical studies. OMVs are naturally decorated with a number of MAMPs, which confer them a potent Th1-skewed adjuvanticity. The local release of IFNγ, IL12 and TNFα is the type of response needed to promote cytotoxic T cell production and to activate phagocytic cells and NK cells. Moreover, *E. coli* OMVs induce immunogenic death of cancer cells, thus facilitating the up-take of tumor antigens by APCs and the production of effector T cells. Finally, the production process of OMVs involves a simple bacterial fermentation, and OMVs are purified by tangential flow filtration of the supernatant with a yield of 50–100 mg/Lt. Assuming that a full therapy would involve a few injections of 5–20 μg of OMVs, 1 Lt fermentation would be sufficient to treat several hundreds of patients.

The most indicated cancers for ISV are obviously those that are easily accessible, such as melanoma, head and neck cancers, breast cancer and lymphomas. They represent approximately one-third of all cancer cases in the USA [24]. However, with the current clinical technologies, all tumors can be treated with ISV. Indeed, most oncological units are able reach any tumor lesion with a 1 mm needle to establish a cancer diagnosis, and the same modalities can be used for in situ vaccination.

One of the foreseeable opportunities with high therapy potential is performing ISV on a primary tumor before surgery [24]. This could be performed as neoadjuvant therapy with no interference with the current standard of care, a key prerequisite to ease regulatory approval. Pre-surgery ISV would offer at least two major advantages. First, the shrinkage of the tumor mass would make the surgical intervention less invasive. Second, it would facilitate the proper activation of the immune responses before tumor removal. Immunologically speaking, it makes little sense to try to force the immune system to recognize and kill cancer cells, and eventually prevent tumor rebound and metastasis formation, if the major source of tumor antigens has been removed.

## Figures and Tables

**Figure 1 cancers-15-03328-f001:**
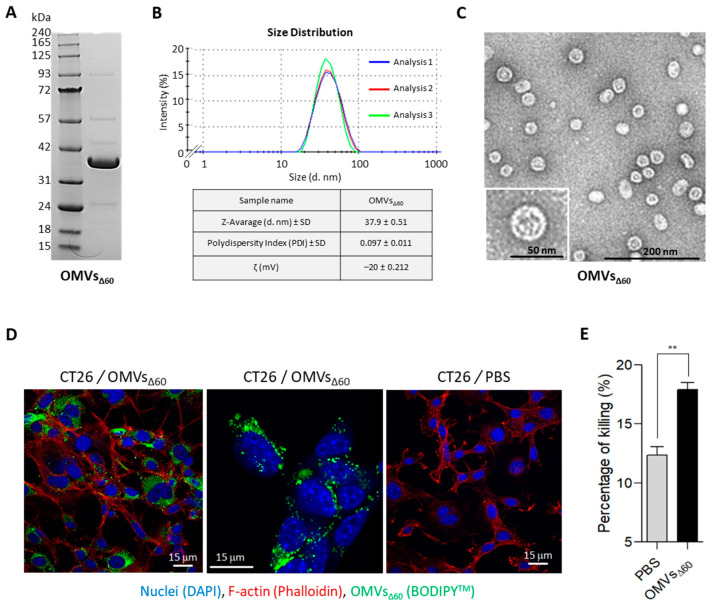
OMV characterization and evaluation of the capacity of OMVs to interact with tumor cells and measurement of the killing properties analyzed by LDH assay—(**A**–**C**) OMV batch preparation and characterization. OMVs_Δ60_ were purified from 1 L culture and concentrated by TFF and dialyzed with sterile PBS. In order to confirm the quality of the preparation and the homogeneity of the batch, the vesicles were analyzed by SDS-PAGE, DLS and Electron Microscopy (EM). More in detail, 10 μg of OMVs was loaded into a 4–12% SDS-PAGE gel and the total protein content visualized by Comassie Blue staining (**A**). Then, size distribution profile of OMVs was determined by DLS using Malvern Zetasizer Nano (NanoZS90, Malvern, UK). The analysis was performed in triplicate and each measurement was reported in the size distribution graph (Blue, Red and Green curves) (**B**). Finally, 5 μL of the OMV batch diluted at 80 ng/μL in PBS was analyzed using a Tecnai G2 Spirit Transmission Electron Microscope (**C**). (**D**) Analysis of OMVs_Δ60_ internalization in CT26 tumor cells by confocal microscopy. Cells were treated with either labeled BODIPY^TM^ 493-503 OMVs_Δ60_ (green) or PBS (as negative control), and cells were stained with DAPI for nuclei detection (reported in blue) and Phalloidin to stain F-actin (red). (**E**) The LDH assay was performed on CT26 tumor cells incubated for 24 h with either PBS or OMVs_Δ60_, to identify the percentage of killing. Statistical analysis was performed using Student’s *t* test (two-tailed). ** *p* ≤ 0.01. Error bars: mean ± SD.

**Figure 2 cancers-15-03328-f002:**
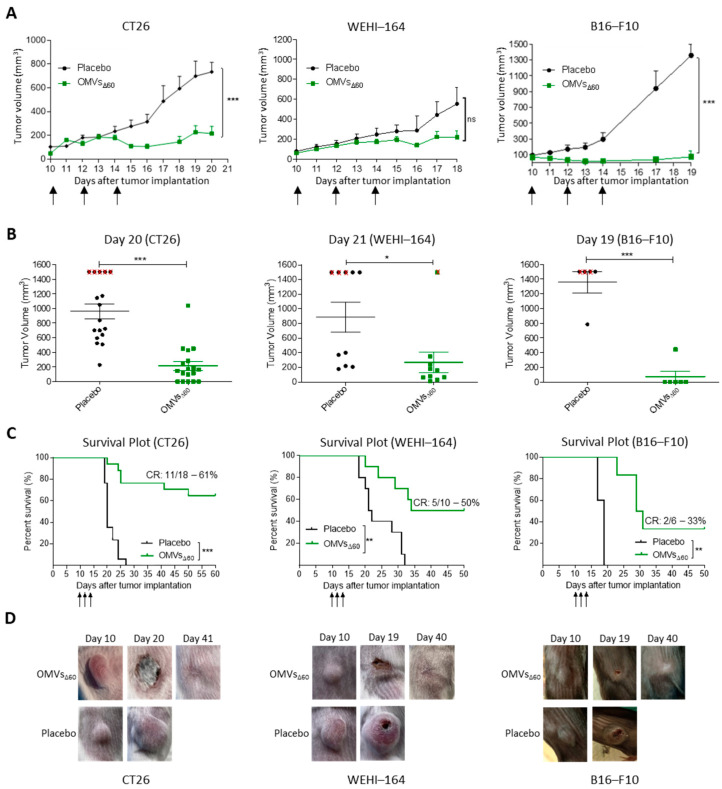
Tumor inhibition and survival analysis induced by OMVs_Δ60_ treatment—(**A**) Therapy study was performed on BALB/c mice bearing colon carcinoma tumors CT26 (left), fibrosarcoma tumors WEHI-164 (middle) or C57BL/6 mice bearing melanoma B16-F10 tumors (right). At day 10, after subcutaneously tumor implantation in the right flank, when tumors reached a volume around 100 mm^3^, treatments started. Injections of 10 μg OMVs_Δ60_ in 50 μL PBS, or PBS alone as a placebo (50 μL), were administered intratumorally every 48 h for 3 times (black arrows). Each line represents means and SEM for each group of mice. Statistical analysis: unpaired, two-tailed Student’s *t*-test (CT26: Placebo n = 17, OMV_Δ60_ n = 18; WEHI-164: Placebo n = 10, OMVs_Δ60_ n = 10; B16-F10: Placebo n = 5, OMVs_Δ60_ n = 6). (**B**) Tumor volume is expressed as single mouse dot-plot at day 20 for the CT26 therapy (left), day 21 for the WEHI-164 (middle) and day 19 for the B16-F10 therapy (right). Red crosses represent sacrificed mice for ethical reasons (tumor volume ≥ 1500 mm^3^). Mice from Placebo group are represented as black circles while OMV immunized mice as green squares. Statistical significance was assessed using two-tailed Student’s *t* test. (**C**) Kaplan–Meier curves of BALB/c mice challenged with CT26 cells (left), WEHI-164 (middle) and C57BL/6 mice challenged with B16-F10 cells (right). Tumor-free mice survival was followed for 50–60 days after the tumor challenge. Black arrows represent the immunization days. Statistical analysis was performed with the comparison of survival curves using Gehan–Breslow–Wilcoxon Test. (**D**) Representative pictures of tumor-bearing mice (CT26 left, WEHI-164 middle and B16-F10 right) treated with OMV_Δ60_ taken before the first treatment (Day 10), during the necrotic and wound healing phase (Day 19–20) and completely cured mice after day 40; compared with placebo-receiving mice. * *p* ≤ 0.05; ** *p* ≤ 0.01; *** *p* ≤ 0.001.

**Figure 3 cancers-15-03328-f003:**
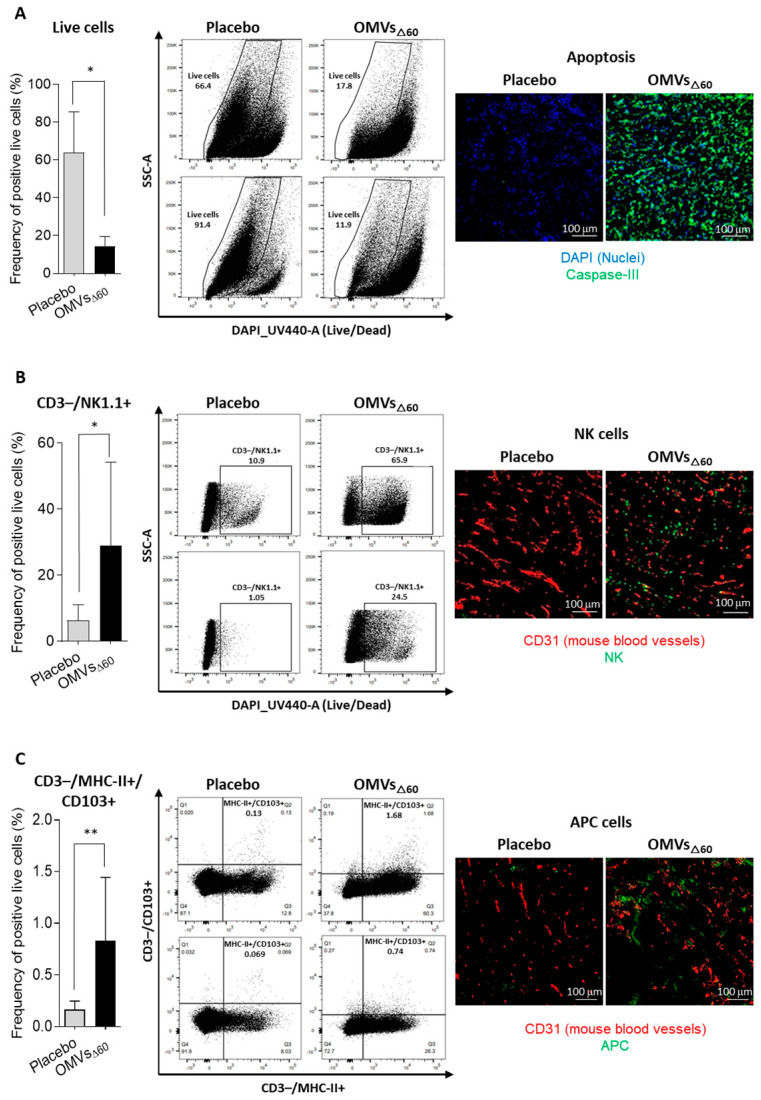
Analysis of Tumor-Infiltrating Lymphocytes (TILs) and cell killing by flow cytometry and immunofluorescence—24 h after one treatment with PBS or 10 μg of OMV_Δ60_, tumors were collected and analyzed by flow cytometry and IF. The graphs in the figure reports the flow cytometry analysis of the frequency of single live cells in tumors treated with either PBS (n = 4) or with OMVs_Δ60_ (n = 4), two plots per immunization group were also represented in the figure. Statistical analysis was performed using Student’s *t* test (two-tailed). * *p* ≤ 0.05, ** *p* ≤ 0.01. Error bars: mean ± SD. (**A**) Frequency of live cells analyzed by FACS (left) and tumor cryosections stained for apoptotic activation marker (right). Blue = Nuclei (DAPI), Green = Casp3. (**B**) Frequency of CD3-/NK1.1+ cells analyzed by FACS (left) and tumor cryosections stained for Natural Killer cell marker (right). Red = mouse blood vessels (CD31), Green = NKp46. (**C**) Frequency of CD3-/MHCII+/CD103+ cells analyzed by FACS (left) and tumor cryosection stained for activated antigen-presenting cells marker (right). Red = mouse blood vessels; Green = DC 33D1. Immunofluorescence: 20× magnification with scale bar 100 μm.

**Figure 4 cancers-15-03328-f004:**
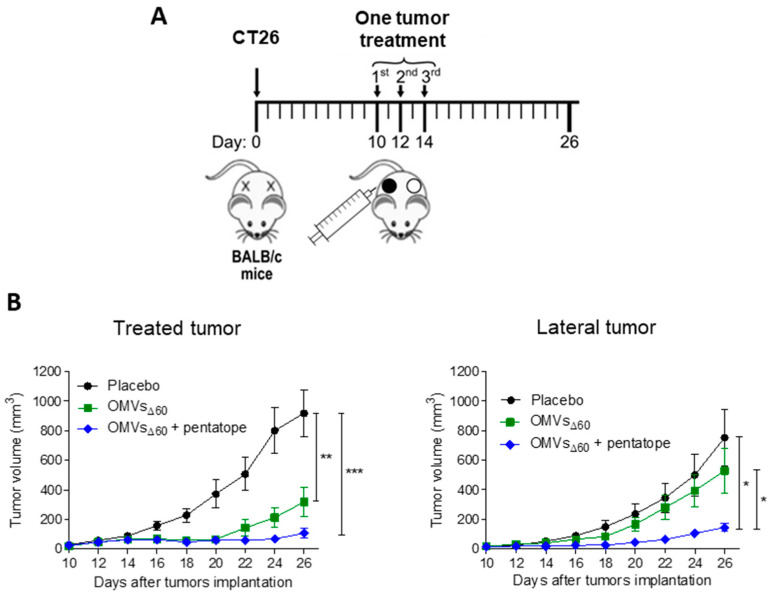
Tumor inhibition by OMVs_Δ60_ ISV in a two-tumor mouse model. (**A**) BALB/c mice were injected s.c. with 2.5 × 10^5^ CT26 cells in both flanks, and the first of the two tumors to reach a palpable size of 50–100 mm^3^ was intratumorally vaccinated three times, two days apart. Mice received either PBS, OMVs_Δ60_, or OMVs_Δ60_ + pentatope (a mixture of five mutation-derived, tumor-specific T cell neo-epitopes). (**B**) Tumor growth was monitored on both sides over a period of 26 days. The figure shows data from two (OMVs_Δ60_, or OMVs_Δ60_ + pentatope) or three (PBS control) independent experiments using groups of five mice each. Statistical significance was assessed using two-tailed Student’s *t* test. Error bars: mean ± SEM. * *p* ≤ 0.05, ** *p* ≤ 0.01, *** *p* ≤ 0.001.

## Data Availability

The data that support the findings of this study are available from the corresponding author upon reasonable request.

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
