# Peer review of "Anti-Tumor Efficacy of In Situ Vaccination Using Bacterial Outer Membrane Vesicles"

_cancers, 2023, doi:10.3390/cancers15133328_

Round 1
Reviewer 1 Report
The authors of this article explored anti-tumor effects and mechanisms of bacterial outer membrane vesicles (OMVs) mediated in situ vaccination in multiple tumor models. The novelty and significance of this work are not well justified in the introduction part. The anti-tumor results are encouraging, but the data presented are not adequate to support the conclusions and the research on related mechanisms is not in-depth enough (some points but not all are listed below). This manuscript is not suitable for publication in Cancers.
(1) The authors did not characterize the size, structure, and components of OMVs.
(2) In Figure 1, the validation experiment of tumor cells internalizing OMVs lacks quantitative data support. The possible mechanism by which OMVs kill tumor cells is not investigated at all.
(3) The author stated that OMVs had a significant inhibitory effect on all validated tumor models. However, the presented data in Figure 2A showed no statistical difference in WEHI-164 tumor-model.
(4) In Figure 3, the author carried out flow cytometry analysis on the inflammatory reaction triggered by intratumoral injection of OMVs. However, the presented data did not show the frequency of typical inflammatory cells (e.g., neutrophils and monocytes). In addition, only two mice were used in the experiment, which lacked reliability.
English writing of this paper is pretty good but the authors need to imporve the scientific rigor.
Author Response
- The authors did not characterize the size, structure, and components of OMVs.
Such characterization was thoroughly described in our work published in 2021 on Journal Extracellular Vesicles (Zanella et al., 2021). In that paper we also reported the mass spectrometry analysis of OMVsD60, which allowed us to define the proteome of the vesicles. Therefore, we believe that reporting the characterization of OMVsD60 would be redundant. However, to satisfy the request from this Reviewer, we have modified Figure 1, including SDS-PAGE, size distribution and the Electron Microscopy analyses of the OMVsD60 lot used in our experiments. These analyses are consistent with what previously published.
(2) In Figure 1, the validation experiment of tumor cells internalizing OMVs lacks quantitative data support. The possible mechanism by which OMVs kill tumor cells is not investigated at all.
We wish to point out that the cytotoxic activity of OMVs from different bacteria (including E. coli BL21) was reported for the first time by Vanaja et. al. in their Cell paper published in 2016. In that paper, they described in detail the killing assay used to demonstrate OMV cytotoxicity on different cells and they thoroughly characterized the mechanisms of OMV cytotoxicity. Therefore, we have essentially followed the protocol reported by Vanaja et al. to confirm that our OMVs, purified from a derivative of E. coli BL21, exerted a similar cytotoxic activity on CT26, the cancer cells we use in our mouse model. The optimization of the assay was out of the scope of our work and we believe there is no reasons to repeat experiments to elucidate mechanisms that have already been well described.
To address the Reviewer’s concern, in the Results section, where the Vanaja’s paper is cited, we have included a paragraph summarizing the published results on OMV internalization and the mechanisms of OMVs cytotoxicity. Moreover, we have analyzed our confocal microscopy images and we estimated the percentage of cells that showed the presence of OMV-associated fluorescence in their cytoplasmic compartment. Such percentage, indicated in the text (approximately 50%), is in line with what published by Vanaja and coworkers.
(3) The author stated that OMVs had a significant inhibitory effect on all validated tumor models. However, the presented data in Figure 2A showed no statistical difference in WEHI-164 tumor-model.
In figure 2A we compare the kinetics of tumor growth up to the time the control and the vaccinated groups are homogeneous (same number of animals). In the WEHI-164 tumor model, between day 18 to day 21 (the days the Animal Facility were asked to measure tumor volumes) the tumors rapidly developed in some of the animals in the control group and these animals were sacrificed because they reached the threshold tumor volume (1.500 mm3) imposed by the Animal Welfare Committee. If we would calculate the statistical difference at day 21, including the sacrificed mice and assigning to them a tumor volume of 1.500 mm3, such difference would become significant (see attached Figure). This said, the effectiveness of OMV administration in the WEHI-164 tumor model can be well appreciated by looking at the data shown in Figure 2B and 2C.
(4) In Figure 3, the author carried out flow cytometry analysis on the inflammatory reaction triggered by intratumoral injection of OMVs. However, the presented data did not show the frequency of typical inflammatory cells (e.g., neutrophils and monocytes). In addition, only two mice were used in the experiment, which lacked reliability.
We agree that the analysis of the tumors in two animals is not sufficient. To this aim, we have extended this analysis of live/dead cells, NKs infiltration and DCs infiltration to four tumors, from both the control and vaccinated groups and we have modified the graphs in Figure 3 to incorporate these results. We also included in the graphs the statistical differences between the two groups.
Regarding the analysis of tumor infiltrating neutrophils and/or monocytes, we agree with this reviewer that such analysis would further support the evidence that OMVs injection triggers an inflammatory reaction in the tumor microenvironment. The reason why we focused our attention on DCs and NKs is because such cell populations play particularly important roles in tumor inhibition. Since the analysis of other cell types in tumor would require the utilization of additional animals, for humanitarian reasons we would prefer not to carry out these experiments which, although interesting, would not add too much to the message of the paper.

Reviewer 2 Report
This article evaluates the value of Bacterial Outer Membrane Vesicles from E. coli BL21 (DE3) Δ60 in ISV in cancer immunotherapy. The results demonstrated tumor inhibition correlates with a rapid infiltration of DCs and NK cells. The date from this study support OMV-based ISV has the potential to become a new treatment strategy of solid tumors. The design of this study is reasonable, the experimental methods are appropriate, and the data are abundant. The research results have certain significance for cancer treatment.
Please modify the following deficiencies as appropriate:
1. In Result 1, the author has the following description: Further incubation resulted in the death of approximately 10% of the cells, as judged by the in vitro ADPH release. Can the authors give a precise time when cell death was observed? Whether the effect of the vaccine on the cells is related to incubation time?
2. The data presented in Figure 1B does not match the results described in the text (10% of the cells were killed?). The number of dead cells in the two groups should be given in the text respectively. Fig 1B should be inserted at the appropriate place in the text.
3.Figure 2C and D should be named in the order in which they appear in the text, with C appearing first and D following.
4. Please correct two errors in this manuscript. In results 3.1: “ADPH release” should be “LDH release”. In results 3.3: “ pentatome” should be “ pentatope”.
Author Response
Please modify the following deficiencies as appropriate:
- In Result 1, the author has the following description: Further incubation resulted in the death of approximately 10% of the cells, as judged by the in vitro ADPH release. Can the authors give a precise time when cell death was observed? Whether the effect of the vaccine on the cells is related to incubation time?
The killing activity was measured at 24 hours and this was clarified in the Results section at page 6.
We did not measure the killing activity of OMVs on CT26 cells as a function of incubation time since we decided to follow the protocol described by Vanaja et al. (in their Cell paper the authors used an incubation time of 17 hours). In that paper, they describe in detail the killing assay used to demonstrate OMV cytotoxicity on different cells and they thoroughly characterize the mechanisms of OMV cytotoxicity. Therefore, we have essentially followed their protocol just to confirm that our OMVs exert a cytotoxic activity on CT26, the cancer cells we use in our mouse model. The optimization of the assay was out of the scope of our work. However, while setting-up the killing assay using HeLa cells, we analyzed the killing activity of OMVs at 24 and 48 hours and we observed a time-dependent killing (please see the attached figure).
- The data presented in Figure 1B does not match the results described in the text (10% of the cells were killed?). The number of dead cells in the two groups should be given in the text respectively. Fig 1B should be inserted at the appropriate place in the text.
We have clarified this point in the Results section, page 6, where we also make reference to Figure 1B (that has become 1F because of the addition of other panels requested by Reviewer 1).
3.Figure 2C and D should be named in the order in which they appear in the text, with C appearing first and D following.
Thank you. Done.
- Please correct two errors in this manuscript. In results 3.1: “ADPH release” should be “LDH release”. In results 3.3: “ pentatome” should be “ pentatope”.
Thank you. Done.

Round 2
Reviewer 1 Report
The authors have extensively revised the manuscript and addressed the concerns from the reviewer; thus, it is suitable for publication.